# Structural and Functional Changes in Soybean Protein via Remote Plasma Treatments

**DOI:** 10.3390/molecules28093882

**Published:** 2023-05-04

**Authors:** Hyun-Joo Kim, Jin Hee Bae, Seonmin Lee, Jinwoo Kim, Samooel Jung, Cheorun Jo, Jin Young Lee, Jung Hyun Seo, Sanghoo Park

**Affiliations:** 1Department of Central Area Crop Science, National Institute of Crop Science, Rural Development Administration, Suwon 16613, Republic of Korea; 2Department of Nuclear and Quantum Engineering, Korea Advanced Institute of Science and Technology (KAIST), Daejeon 34141, Republic of Korea; 3Division of Animal and Dairy Science, Chungnam National University, Daejeon 34134, Republic of Korea; 4Department of Agricultural Biotechnology, Research Institute of Agriculture and Life Science and Technology, Seoul National University, Seoul 08826, Republic of Korea; 5Department of Southern Area Crop Science, National Institute of Crop Science, Rural Development Administration, Miryang 50424, Republic of Korea

**Keywords:** cold plasma treatment, air discharge, ozone, nitrogen oxides, soybean, protein

## Abstract

To the best of our knowledge, few studies have utilized cold plasma to improve soybean protein extraction yield and the functional properties of soybean protein. In this study, we aimed to assess the benefits of remote plasma treatments on soybean with respect to the utilization of soybean protein. This study involved two different sample forms (whole and crushed beans), two different plasma chemistry modes (ozone and nitrogen oxides [NO*_x_* = NO + NO_2_]), and a novel pressure-swing reactor. Crushed soybeans were significantly affected by NO*_x_*-mode plasma treatment. Crushed soybeans treated with NO*_x_*-mode plasma had the best outcomes, wherein the protein extraction yield increased from 31.64% in the control to 37.90% after plasma treatment. The water binding capacity (205.50%) and oil absorption capacity (267.67%) of plasma-treated soybeans increased to 190.88% and 246.23 % of the control, respectively. The emulsifying activity and emulsion stability slightly increased compared to those of the control. The secondary structure and surface hydrophobicity were altered. The remote plasma treatment of crushed soybeans increased soybean protein extraction yield compared to plasma-treated whole beans as well as untreated beans and altered the structural and physicochemical properties of soybean proteins.

## 1. Introduction

Soybean (*Glycine max* L.) is a popular plant product used in human and animal diets, owing to its high protein (approximately 40%) and oil contents (18–22%). Soybean proteins are used in animal and human foods in several forms, such as flour, protein concentrates and isolates, and textured fibers [1]. The proteins stored in soybeans include 7S (β-conglycinin) and 11S (glycinin). The three subunits of soybean β-conglycinin have good conformational flexibility in the quaternary structure [2]. However, the compact quaternary structure of glycinin is generally stabilized through electrostatic and hydrophobic interactions and disulfide bonds, which result in decreased molecular flexibility and relatively poor emulsification properties [3]. Therefore, for the extensive applicability of soybean protein in food formulations, its functional properties, such as emulsion and dynamic surface properties, can be improved through physical and chemical modifications [4].

Soybean protein has been a predominantly marketed plant-based protein for several years. The alkali extraction of soybean protein has been used to prepare the isolate from soy meal (soy flakes or flour). However, the majority of the protein from the meal remains unextracted using this method [5]. Many physical modifications, such as extrusion, ultrasound, microwave, and high-pressure processing, have been applied to soymeal to improve protein extraction yields [6]. To utilize soybean protein as an alternative protein source, further research is needed on the development of novel, eco-friendly technologies that can enhance the functional properties of proteins while improving the protein extraction yield.

Cold plasma (CP) treatment is an emerging technology in the field of nonthermal food processing. As the gas is partially ionized via electrical discharge, CP consists of electrons, ions, neutrals, and free radicals that are not in thermal equilibrium; therefore, it is suitable for analyzing heat-sensitive materials. Since abundant reactive chemical species can be produced in CP, particularly in atmospheric-pressure CPs, they demonstrate bactericidal, fungicidal, and virucidal effects. To improve the applicability of CP in certain foods and agricultural areas, CP-treated water, which is a mixture of reactive oxygen and nitrogen species originating from the plasma, has been actively used. Depending on the plasma characteristics, CP-treated water shows significant differences in the density of reactive oxygen and nitrogen species, pH, oxidation–reduction potential, and electrical conductivity, which can strongly affect seed germination and plant growth [7]. CP improves the functional properties of soybean protein [8] through the inactivation of enzymes and modification of plant proteins. Specifically, to utilize CP for the industrial development of protein materials, it is essential to have research results that apply various factors, such as the form of the sample and CP conditions. However, to the best of our knowledge, very few experiments have attempted to utilize CP to improve the protein extraction yield and functional properties of soybean.

The aims of this study were to (1) evaluate the effects of CP treatment conditions on the protein extraction yield of soybean and (2) explore the effects of CP treatment conditions on the functional properties of soybean proteins.

## 2. Results and Discussion

### 2.1. Ozone and NO_2_ Concentrations in the Pressure-Swing Reactor

Figure 1a,b presents the absolute number densities of ozone (O_3_) and nitrogen oxide (NO_2)_ in the O_3_-mode and NO*_x_*-mode pressure-swing (PS) reactors, respectively. For each species, the characteristics of whole beans are depicted using black scatters and those of crushed beans using red scatters. Notably, NO and NO_2_ were not detectable in the O_3_-mode PS reactor, while O_3_ was not observed in the nitrogen oxide (NO*_x_*) mode. Although NO was mainly produced in the NO*_x_*-mode plasma reactor, it was not detected in the PS reactor, which implies that NO was completely oxidized to NO_2_ during the transport and pressurization process. Time variations in O_3_ and NO_2_ in the PS reactor were consistent with the gas pressure data (Figure 1c,d). During the pressurization period of approximately 300 s, O_3_ and NO + NO_2_ concentrations in each PS reactor were equalized with those in the plasma reactor, reaching (4.5–7.3) × 10^15^ cm^−3^ and (7.3–8.8) × 10^15^ cm^−3^, respectively. We assume that our PS process in the system enabled fast and uniform reactions of highly concentrated O_3_ and NO_2_ with the samples.

Considering that the surface area of the crushed bean is considerably larger than that of the whole bean, the chemical species should be more adsorbed on the crushed bean. Herein, increased adsorption of the reactive species on the crushed beans was clearly demonstrated, since the number densities of O_3_ and NO_2_ were lower in the crushed beans than in the whole beans (Figure 1a,b). Moreover, the time-averaged number density of O_3_ with both types of soybeans decreased during the second pressurization cycle, thereby indicating that O_3_ treatment makes the raw bean more reactive to O_3_ (Figure 1e). In the NO*_x_* mode (Figure 1f), the time-averaged number density of NO_2_ molecules remained almost constant, regardless of the number of cycles.

### 2.2. Functional Properties

Physical treatments, such as ultrasonication, induced cavitation, and microstreaming, result in the breakdown of particles into smaller ones. This increases cell swelling and hydration, thereby releasing proteins into the extracting solvent [9]. However, to the best of our knowledge, studies on the improvement in soybean protein extraction yield under CP conditions have not yet been reported. The protein extraction yield from soybeans using CP is shown in Figure 2a. The protein extraction yield of the untreated sample (whole) was 31.64%, which increased on using CP. Crush-NO*_x_* showed the highest protein extraction yield of 37.90%. The extraction yield of functional components, such as sugars and polyphenols in natural products, increases with CP. Won et al. [10] showed an increase in the polyphenol concentration of mandarin peel after 10 min exposure to microwave-powered nitrogen plasma at 900 W. Rashid et al. [11] highlighted the critical role of CP in enhancing galactomannan extraction, including the generation of reactive species in the extracting solution and modification of the seed surface microstructure. Therefore, the extraction efficiency of functional components differs depending on various factors, such as sample condition, extraction solvent, and radical type during CP treatment.

The water binding capacity (WBC) measured in CP-exposed soybean protein was significantly higher than that in the unexposed controls. When the crushed soybeans were treated with CP in NO*_x_* mode, the WBC count was the highest (Table 1). Additionally, CP effectively increases the hydrophilicity of the food surface [12]. The unfolding of protein chains, placement of more hydrophilic groups on the surface, breakdown of long amyloid chains, and formation of lower-molecular weight components increase the water absorption of macromolecules [13]. Surface oxidation of granules can also occur when in contact with generated reactive species, which can increase surface charge and facilitate the placement of polar functional groups on the granule surface, thereby resulting in increased water absorption. Additionally, protein depolymerization can increase the surface-to-volume ratio of polymer particles, thereby making them more reactive with water molecules [14]. Sharifian et al. [15] showed that the WBC of myofibrillar proteins increases from 21.8% in the control to 92.6% after CP treatment. They suggested that surface modification may have led to the unfolding of proteins to form crosslinks between protein strands, which trap high amounts of water, thereby increasing the WBC.

Oil absorption capacity (OAC) is an important functional property of foods such as pulse flours and proteins. It influences the flavor, texture, and product yield. It is also critical in the manufacturing of products such as doughnuts, pancakes, baked goods, desserts, confectioneries, beverages, meat extenders, and meat analogs [16]. The OAC measured in CP-exposed soybean protein was significantly higher than that in unexposed controls. Similar to the WBC, crush-NO*_x_* had the highest OAC among all the treatments (Table 1). The majority of the changes in OAC have been attributed to protein modifications during CP treatment, revealing the surface and structural modification of the protein and fiber matrix in comparison with starch molecules [17]. This indicates the exposure of nonpolar groups in proteins and an increase in the surface hydrophobicity of flour particles, which is similar to the findings of Bußler et al. [18], who reported that CP increased the WBC and OAC, depending on the CP treatment time and protein composition.

Proteins and lipids generally interact in various food systems, thus imparting a remarkably high capacity for proteins to form emulsions [19]. However, plant proteins are mainly globular proteins with low emulsifying power owing to their large molecular size, which limits their flexibility to rapidly adsorb at the oil–water interface [20]. Table 1 presents the emulsifying activity and emulsion stability of soybean protein based on the CP treatment. The emulsification activity and emulsion stability of soybean protein tended to increase marginally with CP, and the values were the highest for crush-NO*_x_*. This increase during CP treatment is due to an increase in the surface hydrophobicity induced by the exposure of the hydrophobic groups of protein molecules [15]. Basak and Annapure [20] observed that the partial unfolding and degradation of protein aggregates induced by CP treatment can lead to an improved emulsifying capacity. Additionally, Sharafodin and Soltanizadeh [8] indicated that increased solubility leads to enhanced protein access to the water–oil interface during the emulsification process. The mechanism underlying the enhanced functional properties, such as protein extraction yield, WBC, OAC, emulsifying activity, and emulsion stability of soybean protein, could be predicted by structural modifications after CP treatment.

### 2.3. Structural Properties

Circular dichroism (CD) spectroscopy was used to characterize the impact of CP on the secondary structural profiles of soybean proteins. The primary structure of soybean protein consists of a β-sheet and random coil (Table 2). When treated with NO*_x_* using CP, the α-helix, β-turn, and random coil content increased in the whole soybean. The β-sheet content was the highest in crush-NO*_x_*. Different reactive species can have varying effects on protein functionality. For instance, Bu et al. [21] found that H_2_O_2_ can improve the whiteness of pea protein isolate with minimal structural changes, whereas O_3_, NO*_x_*, and OH− were found to distort the conformation of proteins. The loss of helical and/or β-sheet segments upon plasma exposure has been observed previously in whey proteins [22], peanuts [23], and wheat flour [17]. Zhang et al. [24] showed that the α-helix content of soybean protein treated with CP gradually decreases, while that of the random coil marginally increases; the proportions of both β-strands and -turns varied. 

The impact of CP treatment on the tertiary structure of soybean protein was evaluated by measuring surface hydrophobicity, which, in turn, was determined using the bound BPB content. BPB binds to hydrophobic groups; thus, surface hydrophobicity reflects the quantity of exposed hydrophobic amino acid residues that can bind BPB [25]. As shown in Figure 2b, all CP treatments increased the bound BPB content except the crush-O_3_ treatment. An increase in surface hydrophobicity of the crushed soybean–NO*_x_* plasma treatment may be attributed to the unfolding and fragmentation of protein, which can increase the exposure of amino acids [26]. Mehr and Koocheki [27] suggested that the increase in the surface hydrophobicity index may be attributed to the dissociation of reversible protein aggregates and protein subunits that occurs via the etching of plasma in the early stages of treatment. Conformational changes in the tertiary structure of proteins, particularly unfolding, increase surface hydrophobicity [28]. Therefore, it was possible to predict that the secondary and tertiary structures of soybean protein changed with CP treatment, and the maximum effect was observed when crushed soybeans were treated with CP under NO*_x_* conditions.

### 2.4. Heatmap and Correlation Analysis

The results of this study are summarized in the normalized heatmap graph shown in Figure 3a. The crush-NO*_x_* treatment produced the highest protein extraction yield and improved functional properties with structural changes, particularly surface hydrophobicity and α-helices. Additionally, the results of correlation analysis between the physicochemical and structural properties of soybean protein according to CP treatment conditions are shown in Figure 3b. WBC was strongly correlated with the α-helix content (*r* = 0.7981, *p* < 0.001) and protein extraction yield (*r* = 0.9505, *p* < 0.001). The protein extraction yields strongly correlated with the α-helix (*r* = 0.8473, *p* < 0.01). The OAC strongly correlated with surface hydrophobicity (*r* = 0.8488, *p* < 0.001). Thus, the improvement in protein extraction yield, WBC, OAC, emulsifying activity, and emulsion stability of soybean protein could be explained by changes in the secondary structures, particularly α-helices, and surface hydrophobicity after CP treatment.

## 3. Materials and Methods

### 3.1. Samples

Soybeans (Daewonkong) were grown at the National Institute of Crop Science, Rural Development Administration, Daegu, Republic of Korea. Soybeans were harvested in 2021 and stored at 4 °C until further analysis. The samples were pulverized before CP treatment using a micro-hammer–cutter mill (Type 3, Culatti Ag., Zürich, Switzerland), and a particle size of 200 μm was used in this study.

### 3.2. Experimental Setup for Remote Plasma Treatments

Figure 4 shows a schematic representation of our experimental apparatus, which consisted of a plasma reactor, a PS reactor, two identical absorption spectroscopic systems, and a power system. The details of our plasma reactor have been described previously [29,30]. A stainless steel chamber with an inner volume of 815 cm^3^ was used as the plasma reactor to produce surface dielectric barrier discharge (sDBD). The sDBD source was made of 70 mm × 70 mm thin-film electrodes attached to each side of a 1 mm thick, 100 mm × 100 mm fused silica plate. The electrodes had two different forms: a plane high-voltage electrode and a perforated ground electrode. The latter surface was opened with 49 patterns of 7 mm × 7 mm rounded squares in a 7 × 7 alignment, thereby forming surface discharges at the open edge of the electrode (Figure 4). A high-voltage amplifier (20/20C; Trek Inc., Medina, NY, USA) combined with a waveform generator (33512 B; Keysight, Santa Rosa, CA, USA) was used to operate the sDBD source with a 6 kV_pp_ amplitude of the voltage waveform at a frequency of 2 kHz for the O_3_ mode or 4 kHz for the NO*_x_* mode. A hot plate (MSH-20D; Daihan Scientific, Wonju, South Korea) below the plasma reactor was set at 150 °C to maintain the reactor temperature. As the sDBD source formed the lid of the plasma reactor, reactive plasma species were produced inside the reactor.

In this study, plasma processing was customized for food crops, and the plasma source was separated from the samples to maintain constant productivity of reactive species and plasma characteristics, while the samples were remotely treated in the PS reactor. The PS reactor equipped with a rotary pump (Uno6; Pfeiffer Vacuum, Aßlar, Germany) was connected to the plasma reactor through a 1 m polytetrafluoroethylene tube and a needle valve (Figure 4). The PS reactor was cylindrical, with an inner diameter and height of 150 mm. Notably, the remote treatments were distinguished by the dominant chemicals, O_3_ or NO*_x_*, in the plasma reactor when starting the first pressurization period of the PS reactor. In addition, the bean samples were distinguished based on whether they were whole or crushed. Hereafter, the experiments are referred to as the whole bean and crushed bean treated with O_3_ or NO*_x_*; for example, “crush-NO*_x_*” refers to crushed beans treated with NO*_x_*-rich plasma.

First, a 90 mm-diameter Pyrex Petri dish with 60 g of beans was placed in the PS reactor. Subsequently, with the needle valve fully closed, the pump was operated to reduce the gas pressure of the PS reactor from 760 to 0.08 Torr. The plasma reactor was operated at atmospheric pressure to produce reactive oxygen and nitrogen species, including O_3_ and NO*_x_*. When the O_3_ or nitric oxide (NO) level in the plasma reactor reached a maximum concentration, the PS reactor was gradually pressurized by closing the valve and partially opening the needle valve. During the pressurization phase, the gas flow between the two reactors transferred highly concentrated reactive chemical species from the plasma reactor to the sample-containing PS reactor. Pressurization in the range of 0.08–700 Torr was set for 5 min. After treatments (pressurization), the PS reactor was pumped down again for 2.5 min with the needle valve closed. This PS process was repeated thrice for each experiment. The gas pressure in the PS reactor was recorded using pressure gauges (626A; MKS Instruments, Andover, MA, USA and TPR280; Pfeiffer Vacuum).

Between the experiments, special care was taken to conduct all runs with the same initial system conditions; for example, the air modified by plasma discharges during the previous experiments was substituted for ambient air. All the experiments in this study were performed using the same devices without any replacement. Notably, there was no aging effect on the sDBD source during all the experiments.

To simultaneously obtain the absolute number density of reactive oxygen and nitrogen species in both reactors without gas sampling, two identical systems were installed for UV–visible absorption spectroscopy (Figure 4). This allowed the real-time analysis of O_3_ and NO concentrations in the PS and plasma reactor. The absorption spectroscopic system was a combination of a deuterium lamp (DH-mini; Ocean Optics, Dunedin, FL, USA) and a spectrometer (Maya 2000 Pro; Ocean Optics). The lamp light was transmitted through an optical fiber (QP400-2-SR; Ocean Optics), and two collimating lenses (74-UV; Ocean Optics) were used to collimate the light into the chamber and spectrometer. All absorption spectra data were automatically recorded every 0.5 s using a computer, and the absolute number densities of chemicals were then calculated based on the Beer–Lambert law. The absorption path lengths of the system were 15 and 26 cm, respectively. In this study, six reactive species (O_3_, NO, NO_2_, N_2_O_5_, HONO, and HONO_2_) were considered for spectroscopic analysis; the two major reactive species, O_3_ and NO_2_, are presented in Figure 2. Thus, we comprehensively evaluated the protein extraction yield and functional properties of soybeans treated with plasma under different conditions, including the sample shape (whole and crushed) and radical type (O_3_ and NO*_x_*).

### 3.3. Protein Extraction

To extract soybean protein, a defatting process was employed. The soybeans were first pulverized using a grinder (HMF-3100S, Hanil, Wonju, Republic of Korea), and thrice the weight of n-hexane (*w*/*w*) was added to the pulverized soybeans. The mixture was then stirred at 25 °C for 6 h. After stirring, the mixture was transferred to a separatory funnel and allowed to stand for 1 h to obtain a precipitate. This precipitate was then dried for 24 h and used as the sample for protein extraction. Protein extraction was performed using a modified method proposed by Cha et al. [31]. Specifically, 200 mL of 0.2% NaOH solution was added to 20 g of the defatted sample and stirred for 2 h. The pH was then adjusted to 4.5 using 1 N HCl solution and stirred for an additional 30 min. The coagulated protein was centrifuged at 12,000× *g* for 10 min, and only the precipitate was collected, washed with distilled water, and lyophilized for use as a sample. The protein extraction yield was calculated based on the weight of the lyophilized sample.

### 3.4. Functional Properties

#### 3.4.1. Water Binding Capacity and Oil Absorption Capacity

The WBC was measured by mixing 1 g of extracted protein with 40 mL of distilled water and stirring for 1 h [32]. The mixture was then centrifuged (Hi-mac CR22N; Koko Holdings Co., Ltd., Tokyo, Japan) at 1500× *g* for 10 min, and the weight of the precipitated powder was measured. The WBC was calculated as the difference between the initial sample weight (g) and the weight of the precipitated sample (g), while being expressed as a percentage of the initial sample weight (g).

The OAC was determined conventionally, according to the method proposed by Lin et al. [33], with modifications. Briefly, 1.0 g of extracted protein was vortexed with 5.0 mL of soybean oil in a 50 mL conical centrifuge tube. The contents were vortexed for 15 s every 5 min for 20 min. The tubes were then centrifuged at 1600*× g* for 25 min. The excess oil was drained, and the centrifuge tubes were weighed. The OAC was expressed as the amount of oil absorbed per gram of sample.

#### 3.4.2. Emulsifying Activity and Emulsion Stability

The emulsifying activity and stability were determined according to the method proposed by Kim and Park [34], with modifications. The sample (0.5 g) was added to 5 mL of distilled water and dispersed for 1 min at 5000 rpm using a homogenizer (IKA Co., Ltd., Seoul, Republic of Korea); 5 mL of soybean oil was added again for dispersion and mixed in the same manner. The emulsifying activity and emulsion stability were measured using the mixed solution formed at this time. The emulsifying activity was measured by centrifuging the mixed solution at 360*× g* for 5 min, and the emulsion stability was measured by heating the emulsion at 80 °C for 30 min, then cooling and centrifuging at 360*× g* for 5 min. The emulsifying activity and emulsion stability of samples were calculated according to the method proposed by Cha et al. [31], as follows:$$\mathrm{Emulsifying}\;\mathrm{activity}\;\left(\%\right)=\frac{\mathrm{Volume}\;\mathrm{in}\;\mathrm{emulsified}\;\mathrm{layer}\;\left(\mathrm{mL}\right)}{\mathrm{Sample}\;\mathrm{valume}\;\left(\mathrm{mL}\right)}\times 100$$
$$\mathrm{Emulsion}\;\mathrm{stability}\;\left(\%\right)=\frac{\mathrm{Volume}\;\mathrm{in}\;\mathrm{emulsified}\;\mathrm{layer}\;\mathrm{after}\;\mathrm{boiling}\;\left(\mathrm{mL}\right)}{\mathrm{Sample}\;\mathrm{volume}\;\left(\mathrm{mL}\right)}\times 100$$

### 3.5. Structural Properties

#### 3.5.1. Secondary Structure

The secondary structures of the samples were determined using a CD spectrometer (Chirascan VX; Applied Photophysics, Leatherhead, UK). The myosin solution was diluted (0.13 mg/mL) in 0.1 M potassium phosphate buffer (pH 7.4) and transferred to a quartz cell with a path length of 1 mm. The secondary structures of the samples were measured from 200 to 260 nm at a scan rate of 100 nm/min at 25 and 98 °C. The compositions of the α-helix, β-sheet, β-turn, and random coil were estimated using CDNN software (version 2.1, Applied Photophysics). The thermal denaturation of the samples was determined by measuring the CD intensity at 222 nm from 25 to 98 °C; the temperature increased at a rate of 5.0 °C/min.

#### 3.5.2. Surface Hydrophobicity

The surface hydrophobicity of the samples was determined using bound bromophenol blue (BPB), as described by Qian et al. [35]. The samples were reacted with BPB, and after diluting the supernatant 10-fold, the absorbance (Abs.) was read at 595 nm (Multiskan GO Microplate Spectrophotometer; Thermo Fisher Scientific, Waltham, MA, USA). A control sample was prepared by reacting BPB with phosphate buffer. The bound BPB was expressed using the following equation:$$\mathrm{Bound}\;\mathrm{BPB}\;\left(\mu g\right)=200\;\mu g\;\times \frac{\mathrm{Abs}.\;\mathrm{of}\;\mathrm{control}\;\mathrm{sample}\;-\mathrm{Abs}.\;\mathrm{of}\;\mathrm{sample}}{\mathrm{Abs}.\;\mathrm{of}\;\mathrm{control}\;\mathrm{sample}}\times 100$$

### 3.6. Statistical Analyses

All data were analyzed and presented as mean ± standard deviation of triplicate measurements (*n* = 3) using SigmaPlot software (version 14.0; Systat Software, San Jose, CA, USA). Duncan’s multiple range test at *p* < 0.05 was used to compare the treatments with SPSS statistical software (version 18.0; SPSS, Inc., Chicago, IL, USA). Heatmap analysis was performed using MetaboAnalyst 5.0 (https://www.metaboanalyst.ca/; accessed on 15 January 2023) [36] with Euclidean distance measurement, the Ward clustering algorithm, and Pearson’s correlation. Pearson’s correlation coefficient [37] test was used to analyze correlation coefficients (*r*) with a distance measurement. Positive and negative correlation coefficients are indicated by red and blue, respectively.

## 4. Conclusions

Remote plasma treatment can potentially allow soybeans to serve as a valuable source of protein materials. The NO*_x_* treatment of crushed soybeans was found to improve the protein extraction yield when compared to whole soybean grain treated or untreated with CP, and also improve the functional properties by inducing structural changes. Our findings suggest that the application of NO*_x_* treatment on crushed soybeans is an efficient method for treating soybeans prior to their use in industrial applications.

## Figures and Tables

**Figure 1 molecules-28-03882-f001:**
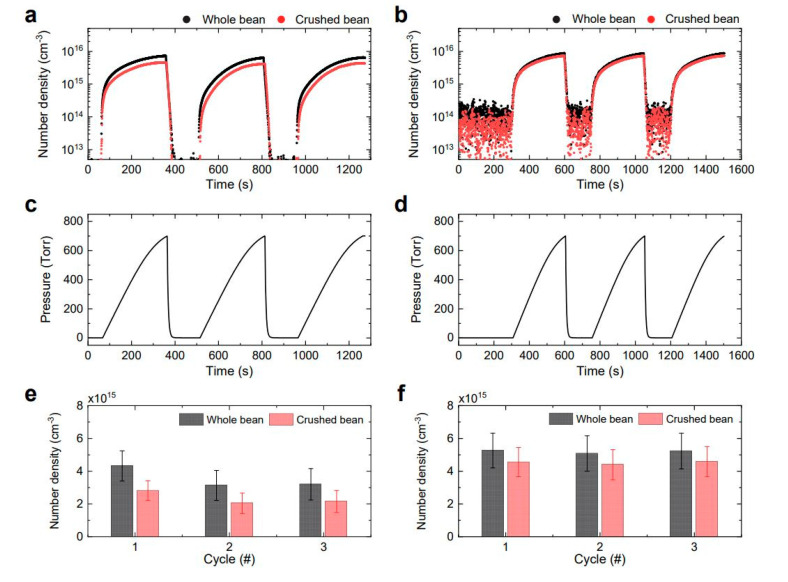
Characteristics of the O_3_−mode and NO*_x_*−mode PS reactors. Time evolution of chemicals, (**a**) O_3_ and (**b**) NO_2_, and (**c**,**d**) gas pressure in each PS reactor. The time−averaged number density of (**e**) O_3_ and (**f**) NO_2_ corresponding to each cycle of O_3_ and NO*_x_* modes, respectively.

**Figure 2 molecules-28-03882-f002:**
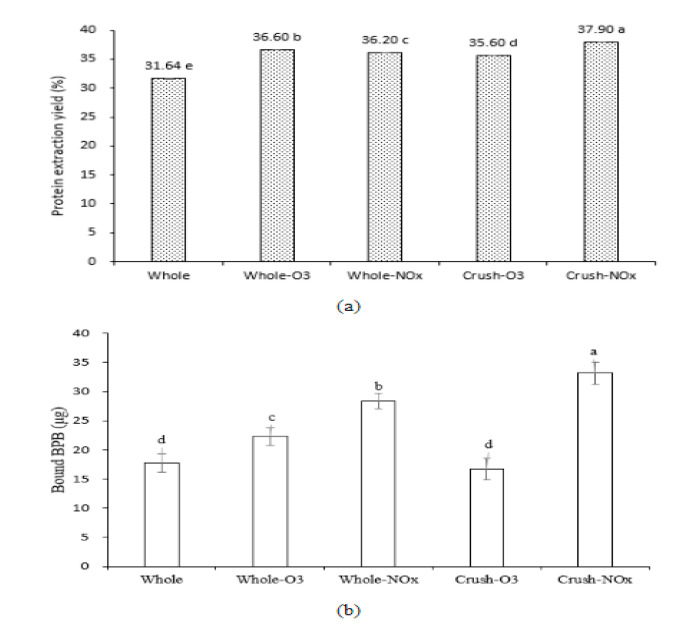
(**a**) Protein extraction yields and (**b**) bound bromophenol blue content of plasma-treated soybean. Different letters in the graph (a–e) indicate a significant difference among treatments according to Duncan’s multiple range test at *p* < 0.05.

**Figure 3 molecules-28-03882-f003:**
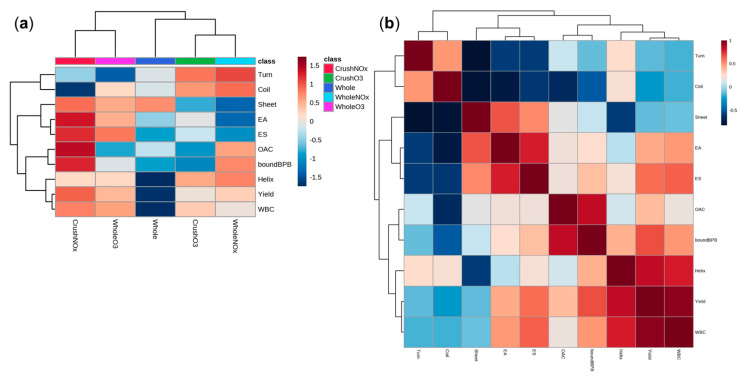
(**a**) Normalized response heatmap and (**b**) Pearson’s correlation coefficient (*r*) matrices of structural and functional properties of soybean proteins treated with atmospheric pressure plasma. The level of individual parameters corresponds to the color scale. A color gradient from blue to red represents a low to high level of the normalized response. Blue and red colors indicate negative and positive correlations between individual parameters, respectively. Abbreviations: Turn, β−turn; Coil, random coil; Sheet, β−sheet; EA, emulsifying activity; ES, emulsion stability; OAC, oil absorption capacity; WBC, water binding capacity; Helix, α−helix.

**Figure 4 molecules-28-03882-f004:**
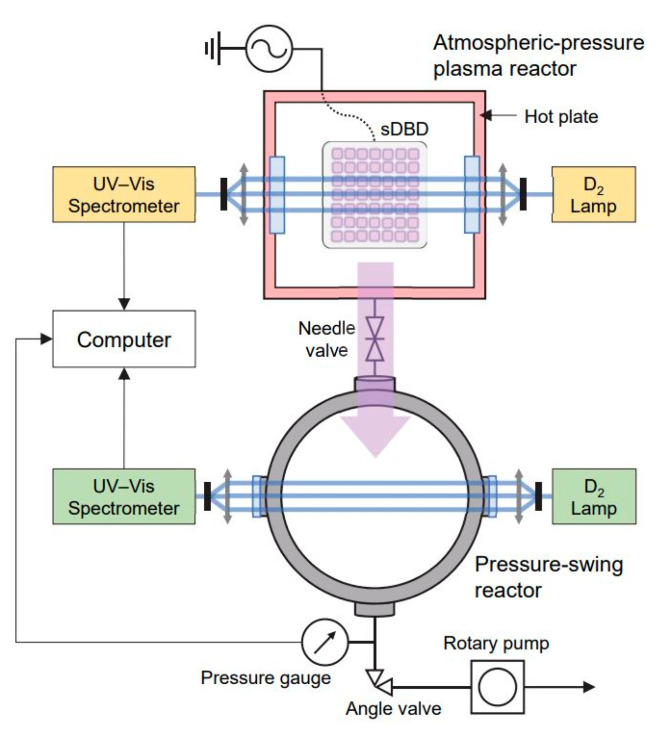
Simplified schematic of the experimental setup for cold plasma (CP) treatments. Samples were treated in the PS reactor, and the detailed processes are described in the main text. Concentrations of O_3_ and NO*_x_* in both the reactors were simultaneously obtained using in situ optical absorption spectroscopic systems.

**Table 1 molecules-28-03882-t001:** Functional properties (%) of plasma-treated soybean proteins.

	Water Binding Capacity	Oil Absorption Capacity	Emulsifying Activity	Emulsion Stability
Whole	190.88 ± 1.11 ^d^	246.23 ± 1.21 ^c^	46.50 ± 1.00 ^bc^	47.33 ± 0.76 ^c^
Whole-O_3_	204.43 ± 1.87 ^ab^	239.23 ± 0.91 ^d^	47.67 ± 0.76 ^ab^	49.50 ± 0.87 ^ab^
Whole-NO*_x_*	200.93 ± 1.52 ^c^	256.67 ± 1.50 ^b^	45.50 ± 0.50 ^c^	47.17 ± 1.04 ^c^
Crush-O_3_	202.57 ± 1.39 ^bc^	237.60 ± 0.61 ^d^	47.00 ± 0.00 ^b^	48.17 ± 0.58 ^bc^
Crush-NO*_x_*	205.50 ± 1.53 ^a^	267.67 ± 0.55 ^a^	48.67 ± 0.29 ^a^	50.00 ± 0.00 ^a^

All data are presented as the mean ± standard deviation of three replicates. Different letters in the same column indicate significant differences between treatments, according to Duncan’s multiple range test at *p* < 0.05.

**Table 2 molecules-28-03882-t002:** Secondary structure compositions (%) of plasma-treated soybean proteins treated with the plasmas.

	α-Helix	β-Sheet	β-Turn	Random Coil
Whole	11.43 ± 0.40 ^c^	36.13 ± 0.76 ^b^	17.93 ± 0.55 ^b^	34.50 ± 1.21 ^d^
Whole-O_3_	17.97 ± 0.60 ^b^	33.23 ± 1.36 ^c^	12.43 ± 1.17 ^d^	36.37 ± 0.76 ^c^
Whole-NO*_x_*	20.47 ± 0.51 ^a^	13.50 ± 0.17 ^e^	25.00 ± 0.36 ^a^	41.03 ± 0.15 ^a^
Crush-O_3_	19.47 ± 0.85 ^a^	17.70 ± 0.53 ^d^	23.40 ± 1.40 ^a^	39.43 ± 0.21 ^b^
Crush-NO*_x_*	17.93 ± 1.12 ^b^	39.47 ± 0.64 ^a^	16.20 ± 0.56 ^c^	26.40 ± 0.53 ^e^

All data are presented as the mean ± standard deviation of three replicates. Different letters in the same column indicate significant differences between treatments, according to Duncan’s multiple range test at *p* < 0.05.

## Data Availability

Not applicable.

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
