# Peer review of "Structural and Functional Changes in Soybean Protein via Remote Plasma Treatments"

_molecules, 2023, doi:10.3390/molecules28093882_

Round 1

Author Response

We thank you and the reviewers very much for the time and effort spent to review our manuscript and provide insightful suggestions and comments.

We have revised the manuscript molecules-2343648 titled " Structural and functional changes in soybean protein via remote plasma treatments” by Kim et al. based on the suggestions and comments.

The changes in the revised manuscript are marked in red-colored font for the convenience of the review process.

Please check the attachment file.

Reviewer 2 Report

Comments to authors:

The title should be "Structural and functional changes in soybean protein via remote plasma treatments", as the properties investigated were more functional attributes than physicochemical properties. Additionally, the quality (resolutions) of Fig. 1, 2 & 3 should be improved. Finally, the abstract should be rewritten, particularly from L23 – 28, to clearly show the difference between the plasma exposed and the control samples.

Lines 33: "18 – 22 %" instead of "18 %–22 %".

Line 50, 63, 69, 103, 144, 170, 207, 220, 315 & 376: "Functional properties" instead of "Physicochemical properties"

Lines 237: Define the word "PS".

Lines 240 – 241: "70 × 70 mm" instead of "70mm × 70 mm".

Lines 241: "100 × 100 mm" instead of "100 mm × 100 mm".

Author Response

(The authors gave the same response as above.)

Round 2
